# The Effect of Alginate/Hyaluronic Acid Proportion on Semi-Interpenetrating Hydrogel Properties for Articular Cartilage Tissue Engineering [note 1]

**DOI:** 10.3390/polym17040528

**Published:** 2025-02-18

**Authors:** Izar Gorroñogoitia, Sheila Olza, Ana Alonso-Varona, Ane Miren Zaldua

**Affiliations:** 1Leartiker S. Coop., 48270 Makina-Xemein, Spain; igorronogoitia@leartiker.com; 2Faculty of Medicine and Nursing, University of the Basque Country (UPV/EHU), 48940 Leioa, Spain; sheila.olza@ehu.eus (S.O.); ana.alonsovarona@ehu.eus (A.A.-V.); 3E2S UPPA, CNRS, IPREM, Universite de Pau et des Pays de l‘Adour, 64600 Anglet, France; 4MANTA-Marine Materials Research Group, E2S UPPA, Universit’e de Pau et des Pays de l’Adour, 64600 Anglet, France

**Keywords:** hydrogel, alginate, hyaluronic acid, bioprinting, scaffold

## Abstract

One of the emergent regenerative treatments for the restoration of the articular cartilage is tissue engineering (TE), in which hydrogels can functionally imitate the extracellular matrix (ECM) of the native tissue and create an optimal microenvironment for the restoration of the defective tissue. Hyaluronic acid (HA) is known for its potential in the field of TE as a regenerative material for many tissues. It is one of the major components of the articular cartilage ECM contributing to cell proliferation and migration. HA is the only non-sulphated glycosaminoglycan (GAG). However, herein, we use a HA presenting a high amount of sulphated glycosaminoglycans (sGAGs), altering the intrinsic properties of the material particularly in terms of biological response. Alginate (Alg) is another polysaccharide widely used in TE that allows stiff and stable hydrogels to be obtained when crosslinked with CaCl_2_. Taking the benefit of the favourable characteristics of each biomaterial, semi-interpenetrating (semi-IPN) hydrogels had been developed by the combination of both materials, in which alginate is gelled, and HA remains uncrosslinked within the hydrogel. Varying the concentration of alginate and HA, the final rheological, viscoelastic, and mechanical properties of the hydrogel can be tailored, always seeking a trade-off between biological and physico-mechanical properties. All developed semi-IPN hydrogels have great potential for biomedical applications.

## 1. Introduction

Nowadays there are several clinical treatment methods for articular cartilage damage, which include palliative, reparative, and regenerative treatments. The first one reduces the clinical symptoms (local pain) but healing of the tissue does not happen. In reparative treatments, a fibrocartilage is formed rather than the hyaline cartilage, leading to misfunction and a lack of adequate mechanical properties of the tissue. Among the regenerative treatments, autologous chondrocyte implantation (ACI) and matrix-assisted autologous chondrocyte implantation (MACI) are the most known and effective ones so far, in which isolated autologous chondrocytes are cultured in vitro and injected into the damaged site [1,2]. A new approach that can further achieve articular cartilage restoration satisfactorily is tissue engineering (TE), where scaffolds, cells, and signalling factors are involved [2,3,4,5]. Scaffolds can imitate the extracellular matrix (ECM) of native tissue, both structurally and functionally, and provide the suitable and supportive 3D microenvironment for the regeneration of defective tissue. The scaffold should be biocompatible and biodegradable and should present a porous structure that permits cell attachment, proliferation, and the maintenance of a differentiated phenotype. In addition, the degradation rate of the scaffold should match that of the new ECM synthesis, and they must have the ability to present similarities to biomechanical properties of native tissue. Scaffolds can be fabricated by 3D bioprinting that allows the fabrication of high precision complex constructs with customized geometries from 3D digital models. The most common technique for tissue fabrication is extrusion bioprinting (EBB), in which living constructs can be manufactured layer by layer with the precise positioning of bioinks [6,7,8,9,10]. 

Among all of the biomaterials available for cartilage regeneration, natural polysaccharide-based ones such as alginate (Alg) and hyaluronic acid (HA) have great potential. Due to their natural origin and abundance, they both are inexhaustible. However, the fact that they come from a natural source makes them have batch-dependent properties [3,11]. Thus, knowing in detail the intrinsic properties of each batch at every moment is essential since they will govern the hydrogel’s final properties. One of the most abundant glycosaminoglycans (GAGs) in the native ECM of cartilage is HA, which plays a crucial role in the structural and functional properties of cartilage. Its major disadvantages, however, are its poor mechanical properties and degradability. HA is involved in the water adsorption and retention, lubrication, and compression bearing of native tissue, and it is known to interact with several cell surface receptors such as CD44 and more [3,12,13,14,15,16,17,18]. It consists of repeating disaccharide units of β-N-acetyl-D glucosamine (Glc-NAc) and D-Glucuronic acid (GlcA) and can be extracted from different sources: bovine vitreous humors, rooster combs, the skin of shark, umbilical cords, and bacteria [12,15]. It is the only non-sulphated GAG and can exhibit very high molecular weight (*M_w_*) [1,10] (*M_w_* affects its viscoelastic and biological properties [12,14,15]), unlike the sulphated GAGs (sGAGs). These sGAGs are characterized by having very low *M_w_* and have key biological properties for cartilage tissue engineering (CTE) like cell recognition and signalling [13,17,19]. Another biopolymer widely used and explored in CTE is alginate [20] since it shows similar structure to GAGs. It is extracted from brown seaweed and is composed of β-D-mannuronic acid (M) and α-L-guluronic acid (G) repeating monomer units. Its chemical structure (*M_w_* and M/G ratio) varies depending on its origin [21,22,23,24,25] and will determine the final properties of the scaffold, as demonstrated in several studies [24,26,27,28] and in our previous work [11]. The gelation of alginate is mediated by divalent ions such as calcium that reacts with carboxyl groups (COO^−^) of G-blocks of alginate, forming a so-called “egg-box” structure [20,21,22,24]. Because of its ease of use, low cost, biocompatibility, versatility, and rapid gelation, robust hydrogels can be obtained. Nevertheless, one of the main drawbacks of alginate is its low bioactivity that could be addressed by chemical modification or combining with other polymers. 

The combination of alginate and hyaluronic acid has demonstrated efficient application in CTE as reported by several authors. Janarthan et al. [29] used a mix of alginate and hyaluronic acid hydrogel to print different structures of various layers by 3D bioprinting, and the in vitro live/dead assay proved the excellent biocompatibility of the constructs using chondrocytes. Nedunchezian et al. [30] showed the chondrogenic capability of adipose-derived stem cells (ADSC) combined with alginate/hyaluronic acid (Alg/HA) hydrogel constructs printed by 3D bioprinting technology, and Antich et al. [31] developed a semi-interpenetrating (semi-IPN) Alg/HA hydrogel that successfully promoted chondrogenesis and maintained the chondrocyte phenotype using chondrocytes. So, these hybrid hydrogels are highly desirable 3D supportive microenvironments for cartilage repair.

The aim of this study is to develop semi-IPN hydrogels of alginate and hyaluronic acid and validate them as potential candidates for use in biomedical applications such as the restoration of articular cartilage injuries. Indeed, we used a HA that came from natural origin (Wharton’s jelly extracted from the umbilical cord) and contained a high amount of sGAGs. This helped us to develop bioactive hydrogels as sGAGs acted as bioactive molecules for providing cell adhesion and proliferation. At the same time, this helped us to mimic the native cartilage even better in terms of composition, thus further approaching the targeting of functional properties. In addition, the alginate used in this work enabled us to develop stiff hydrogels when crosslinked with CaCl_2_. So, exploiting the favourable properties and strengths of each biomaterial, novel Alg/HA hybrid semi-IPN hydrogels were developed for CTE, where alginate was crosslinked by ionic crosslinking and HA remained uncrosslinked within the hydrogel. Finally, to verify these hydrogels as possible candidates for CTE and asses their suitability in 3D bioprinting, rheological, mechanical, and biological characterization were performed, and the printability was evaluated. 

## 2. Materials and Methods

Sodium alginate (Alg), calcium chloride (CaCl_2_, anhydrous), phosphate-buffered saline (PBS), glutaraldehyde, and dimethyl sulfoxide (DMSO) were purchased from Sigma-Aldrich, St. Louis, MO, USA. Minimum essential medium (MEM), Trypsin-EDTA, foetal bovine serum (FBS), penicillin, non-essential amino acids (NEAA), sodium pyruvate (NaPyr), trypan blue stain, Presto Blue stain and propidium iodide (PI) were provided by ThermoFischer Scientific, Waltham, MA, USA. Murine fibroblast cells (L929), which are a commercial cell line, were obtained from American Type Culture Collection (ATCC, Manassas, VA, USA). Hexamethydisilazane was obtained from Electron Microscopy Sciences, Hatfield, PA, USA. Ethanol was obtained from ENMA S.L, Spain, calcein-AM was obtained from Abcam, Cambridge, UK and gentamicin was obtained from Gibco^®^, Cambridge, UK. Native whartonide hyaluronic acid (HA) was provided by Histocell S.L Regenerative Medicine, Derio, Spain. The properties of HA were indicated in the certificate of analysis (Table 1). The molecular weight (*M_w_*) of sodium alginate was determined by POLYMAT, Donostia-San Sebastián, Spain (Table 2). Detailed information about the experimental process of obtaining the *M_w_* of sodium alginate is included in the Appendix A.

### 2.1. Hydrogel Preparation and Crosslinking

Alginate and hyaluronic acid solutions as single components were prepared at different concentrations (2–4% *w*/*w*) in PBS. Alginate solutions were mixed vigorously under a propeller-type stirrer and using a magnetic stirrer in the case of hyaluronic acid solutions, both at room temperature until complete dissolution. Alginate/hyaluronic acid (Alg/HA) hybrid solutions were prepared by adding alginate powder first to PBS until complete dissolution under a propeller-type stirrer at room temperature and hyaluronic acid powder later at different concentrations (Table 3). The concentration of the formulations was selected based on their viscosity. Higher or lower concentrations of both polymers caused a too high or too low viscosity for bioprinting and processability. As shown in Table 3, the molar ratio of both polymers varied considerably, obtaining formulations with high alginate content in contrast to hyaluronic acid content. All solutions were stored at 2–8 °C to prevent polymer degradation. 

Alginate bulk gels and Alg/HA semi-interpenetrating (semi-IPN) bulk gels were prepared in a Petri dish by adding 100 mM CaCl_2_ solution and were left to crosslink for 24 h. In semi-IPN hydrogels, only alginate was crosslinked while HA remained uncrosslinked. Hydrogels for compression testing and biological characterization were punched with a hole puncher of 6 mm diameter and the ones for viscoelastic measurements using a hole puncher with a diameter of 20 mm. 

### 2.2. Rheological Characterization

The shear viscosity of all hydrogel precursors was measured as a function of shear rate at 23 and 37 °C using a HAAKE MARS III (ThermoFischer Scientific, Waltham, MA, USA) rheometer equipped with a peltier element for temperature control. A plate–plate geometry (20 mm, aluminium, gap = 1 mm) was used for the tests. Shear rate varied from 0.5 to 1000 s^−1^. Flow data were adjusted to the Ostwald–de Waele and Carreau–Yasuda models. Appendix A shows the viscosity-curves of different HA batches. All of the flow parameters are summarized in Appendix A.

### 2.3. Viscoelastic and Mechanical Characterization

Dynamic shear experiments were performed for the determination of the storage modulus (*G’*) of each hydrogel (h: 2–4 mm, D ≈ 20 mm) at 23 and 37 °C (conditioning time = 5 min) using the Peltier module integrated with parallel plates (20 mm, aluminium serrated, *F_n_* = 0.5 N autotension). Time varied from 0 to 300 s, applying a frequency of 1 Hz and a constant deformation in the linear viscoelastic regime (0.01%). 

The mechanical properties of all hydrogels (h: 2–4 mm, D: 5–6 mm) were determined by applying unconfined compression (UC) using Univert (CellScale, Waterloo, ON, Canada) equipped with a 10 N load cell. A pre-load of 0.1 N was first applied to set the zero point and then samples were compressed to 40% of deformation at a rate of 25%/min in air at room temperature and in a PBS bath at 37 °C (conditioning time = 5 min). Tangent modulus at a target strain of 10% was determined for each hydrogel. 

Appendix A provides an example of a time sweep experiment (Appendix A) and a stress–strain curve (Appendix A), as well as all of the experimental data (Appendix A). 

### 2.4. Scaffold Fabrication Process

Printed structures (0.84 × 30 × 30 mm, pore size: 5 mm^2^) were fabricated layer by layer (4 layers) using a BIO V1 3D bioprinter (REGEMAT 3D, Granada, Spain) equipped with three syringes and one fused deposition modelling (FDM) extruder, consisting of hardware and Designer software (REGEMAT 3D, Granada, Spain) that are connected by an electronic control unit (ECU). Scaffolds were printed in glass slides (76 × 52 mm) at room temperature using a 5 mL syringe and a 27G conical nozzle inner diameter. Nozzle speed was set to 5 mm/s, the offset to 0.5 mm, and the flow rate to 0.6 µL/s. 

### 2.5. Printability Evaluation

The printability of fabricated structures was evaluated qualitatively and quantitatively. The latter was performed by measuring the experimental filament diameter (*D_exp_*), area of pores, and perimeter of pores of printed structures. Please find further details about the quantitative evaluation in Appendix A. The qualitative evaluation was carried out by visual 3D studies to check the quality and printability of the fabricated scaffolds (4 layers) and accomplished before and after immersing the scaffolds in 100 mM CaCl_2_ solution. The structures were compared with their homologues to analyse the influence of hyaluronic acid addition in the printability and quality of printed constructs.

### 2.6. Preparation of Cell-Laden Solution and Bioprinting of Constructs

To prepare the cell-laden Alg/HA solution for bioprinting, a 3Alg2HA hydrogel was selected. Briefly, L929 cells were resuspended in 200 µL MEM cell culture medium and were mixed with a 3Alg2HA hybrid solution to a concentration of 1 × 10^6^ cells/mL using a syringe and carefully stirring until obtaining a homogeneous distribution of the cells. The printing process was performed using sterile material. The cell-laden bioink was printed using a BIO V1 3D bioprinter (REGEMAT 3D, Granada, Spain) with a printing conical nozzle of 22G inner diameter and a flow rate of 2 µL/s. The syringe was set to 37 °C and glass slides (76 × 52 mm) at room temperature were used as a printing bed. The offset was set to 1 mm. After printing the constructs, they were crosslinked with 100 mM of CaCl_2_ for 5 h and were cultured in MEM supplemented with 1 mM of CaCl_2_ for 1, 3, and 7 days. Cell-laden 3Alg2HA hydrogels without printing in the bioprinter were used as control samples.

### 2.7. In Vitro Cytotoxicity Assay

In vitro cytotoxicity was determined following the International Standard Organization (ISO) 10993-5:2009 normative [32], using L929 mouse fibroblast cells. As complete medium MEM was used, it was supplemented with 1% *v*/*v* of non-essential amino acids, sodium pyruvate 1 mM, 1% *v*/*v* of penicillin and streptomycin (100 U/mL), and 0.1% *v*/*v* of gentamicin plus a 10% *v*/*v* of bovine foetal serum. The extracted culture medium was prepared under sterile conditions following the ISO 10993-12:2021 [33], immersing each sample of alginate and Alg/HA hydrogel (≈0.2 g, diameter: 6 mm, height: 5 mm) previously sterilized by ultraviolet (UV) light for 30 min, in 1 mL of complete medium for 24 h at 37 °C and 5% CO_2_ in a cell culture cabinet. L929 cells were seeded on a Sarstedt 96-well cell culture plate at a density of 4 × 10^3^ cells/well in complete medium (100 µL) and incubated at 37 °C for 24 h. For the blank, complete medium without cells was used. As a positive control, 10% *v*/*v* of DMSO in complete medium was used and as a negative control, only complete medium with cells was used, which were also incubated at the same conditions as the samples. After the incubation time, the complete medium was removed from the wells and extractive medium and controls (100 µL) were added to the corresponding wells and incubated at 37 °C and 5% CO_2_. The cell viability was determined after 24, 48, and 72 h using the Presto Blue^TM^ (ThermoFischer Scientific, Waltham, MA, USA) cell viability reagent, which was added to the wells and incubated for 3 h before the absorbance measurement at 570 and 600 nm using the microplate reader Synergy HT (Biotek, Shoreline, WA, USA). Cell viability (%) was calculated as follows: (1)$$\mathrm{Viability}\;\left(\%\right)=\;\frac{A_{\mathrm{test}}}{A_{\mathrm{control}}}\;\times \;100$$
where *A_test_* is the absorbance of the sample cells and *A_control_* is the absorbance of the negative control cells. *A* value above 70% was considered without any cytotoxic potential according to ISO 10993-5:2009 [32].

### 2.8. Cell Morphology and Adhesion

The morphology and the adhesion of the L929 cells on the surface of the materials was assessed by a Hitachi S-3400 (Hitachi, Chiyoda, Japan) scanning electron microscopy (SEM). Prior to cell seeding, samples were sterilized with UV for 30 min, pre-wetted in complete medium and incubated at 37 °C for 24 h. Later, hydrogel samples were placed in 24-well ultralow attachment plates (Corning, NY, USA) and cells with a density of 5 × 10^4^ cells/well were seeded on the top surface of each hydrogel. Before acquiring SEM images at the time of 72 h, the samples were washed three times with PBS, fixed with 2% *v*/*v* glutaraldehyde in 0.01 M PBS buffer for 30 min at 4 °C and finally rinsed with PBS. The dehydration of samples was carried out through a rising series of graded aqueous ethanol solutions and hexamethyldisilazane was employed to desiccate them for 10 min. Finally, the samples were coated with a thin layer of gold by sputtering.

### 2.9. Cell Viability of Seeded Cells on Hydrogels and Bioprinted Cells

In order to evaluate the viability of cells adhered to the materials (seeded cells) a live/dead assay was performed. Cells cultured for 7 days on the surface of hydrogels were washed in 1X PBS and incubated with a mix of 4 µM of calcein-AM and 5 µM propidium iodide PBS solution for 20 min at room temperature in dark conditions. The samples were then observed under a fluorescence microscope (ZEISS Apotome 3) to visualize live cells (stained green; *λ_ex-em_* = 495–515 nm) and dead cells (stained red; *λ_ex-em_* = 535–615 nm). 

The viability of cells encapsulated in the bioprinted constructs and control samples (not printed constructs) was analysed following the same procedure, but constructs were cultured for 1, 3, and 7 days. A confocal fluorescence microscope (ZEISS LSM 800, ZEISS, Oberkochen, Germany) was used for sample observation. 

All images were analysed by ImageJ software (1.54k, Wayne Rasband, Bethesda, MD, USA). The quantitative evaluation of seeded cells was carried out by splitting the images into four sections and performing the statistical analysis. All images were quantified using the colour threshold technique and particle counting. The cell viability (%) was calculated as (number of viable cells/total number of cells) × 100.

### 2.10. Statistical Analysis

All data are presented as mean ± standard deviation (SD) (error bars) of at least three sample replicates. Statistical comparisons between the different groups were made by one-way analysis of variance (ANOVA) followed by Tukey’s multiple post-hoc comparisons test and repeated measures ANOVA followed by Dunnett’s multiple comparisons test using Prism^®^ statistical software (9.5.1, GraphPad, San Diego, CA, USA). * *p* < 0.05; ** *p* < 0.01; *** *p* < 0.001; and **** *p* < 0.0001. Values of *p* < 0.05 were statistically significant.

## 3. Results

The intrinsic properties of alginate (Alg) and hyaluronic acid (HA) polymers are shown in Table 1 and Table 2. As mentioned previously, the HA provided by Histocell presented a molecular weight (*M_w_*) of ≤2000 kDa and was composed of sulphated glycosaminoglycans (sGAGs) that will act as bioactive molecules, allowing the interaction with cells and favouring the biological properties of hydrogels. Three different HA batches were used, the first one for the non-biological characterization and the remaining two for the biological characterization. With the aim of minimizing differences in properties between the batches as much as possible, the *M_w_* of 230,002 + 230,005 and 230,029 + 230,056 batches was adjusted to the *M_w_* of the 210,041 batch by combining two different extracts. Please find further details in Appendix A. The sodium alginate exhibited a *M_w_* of 392 kDa allowing us to develop printable and robust hydrogels when gelled with the crosslinking agent calcium chloride (CaCl_2_).

### 3.1. Rheological Characterization

Rheological properties such as shear-thinning ability and fast recovery are essential characteristics for a hydrogel precursor to obtain structures with high shape fidelity in 3D bioprinting applications [34].

The shear viscosity of all hydrogel precursors was determined by rotational shear-test experiments where flow curves were obtained at 23 and 37 °C; in Figure 1, data at 37 °C are presented. From these viscosity curves, Newtonian viscosity (*η*_0_) was determined, whereas *k* and *n* parameters were determined from stress curves). All parameters (*η*_0_, *k*, and *n*) and their obtention are summarized in the Appendix A. 

In relation to single-component Alg and HA solutions, Appendix A shows the Ostwald–de Waele parameters obtained from the power-law model Appendix A and Newtonian viscosity obtained from Carreau–Yasuda model Appendix A for all solutions at different concentrations. It can be observed that in both cases, the higher the concentration of Alg or HA, the higher the zero-shear viscosity (*ƞ*_0_) due to a higher amount of polymer chains dissolved in the medium at high concentrations, resulting in more entanglements among the polymer chains. This effect was much more pronounced on Alg, where a small change in concentration led to a high change in viscosity (10 times higher when concentration was doubled), whereas in the case of HA, viscosity could almost not be modulated with concentration. The zero-shear viscosity of Alg was greater than the HA at all concentrations. Considering that the viscosity is determined by the molecular weight (*M_w_*), HA should present higher viscosity than Alg since it displayed a *M_w_* of ≤2000 kDa (Table 1) and Alg, however, was 392 kDa (Table 2). Nevertheless, HA contained a high amount of sGAGs, triggering a reduction in viscosity due to the very low *M_w_* of these sGAGs compared to HA (non-sulphated). The *M_w_* of the sGAGs can vary from 10 to 50 kDa [13], contributing to the low viscosity obtained. Moreover, the *n* values for HA were close to one which indicated almost a Newtonian behaviour, while Alg exhibited pseudoplastic behaviour (*n* < 1) [35,36], as it can also be observed in Figure 1. This is favourable for bioprinting since it can reduce cell death during the extrusion process where shear forces are produced. The greater the concentration, the more the pseudoplastic behaviour is pronounced. Regarding temperature, zero-shear viscosity decreased slightly at 37 °C (Appendix A), mainly in Alg samples where the temperature effect was greater at higher Alg concentrations due to a possible breaking of polymer entanglements [37,38,39]. The influence of temperature for HA could be considered negligible.

With respect to alginate/hyaluronic acid (Alg/HA) hybrid solutions, the Ostwald–de Waele parameters and Newtonian viscosity are presented in Appendix A. Initially, intermediate zero-shear viscosity values of the hybrid solutions could be expected, however, zero-shear viscosity increased when both materials were mixed, obtaining higher values than Alg (Appendix A and Figure 1). For example, the viscosity of 4Alg2HA went up enough (>100 Pas) that the behaviour of the curve changed and therefore, it did not have a Newtonian zone (see Figure 1C, blue data). This synergetic effect is attributed to interactions between Alg chains and HA chains, which are forming physical entanglements and hydrogen bonds (H bonds) (see Figure 2), causing an increase in viscosity [40,41,42]. Apart from this, the viscosity value increased with increasing both Alg and HA concentrations where the interactions between Alg polymer chains and HA polymer chains were higher. All formulations presented a shear-thinning behaviour (*n* < 1) [35,36], closer to the alginate value than the HA value. This means that the interaction between both polymers is destroyed upon the application of shear and the reduction in viscosity is high. Finally, all mixtures exhibited lower viscosity at higher temperatures (Appendix A), concluding that the temperature energy might break the H bonds [43,44,45,46] and polymer chain entanglements [37,38,39] formed within the biopolymer hybrid solutions, thus increasing the mobility between polymer chains. 

### 3.2. Viscoelastic and Mechanical Characterization

The viscoelastic and mechanical properties of all hydrogels crosslinked with 100 mM of CaCl_2_ at 23 and 37 °C are displayed in Appendix A. It should be noted that dynamic shear tests at 37 °C were performed by heating the lower plate of the rheometer (thermal conduction) whereas the compression tests were carried out using a saline bath (PBS) at 37 °C (thermal convection) with the aim of simulating physiological conditions. Concerning viscoelastic measurements, at 37 °C, a lower storage modulus *G’* was obtained because the thermal energy could be breaking the physical or intermolecular interactions such as chain entanglements and H bonds between Alg and HA polymer chains [46]. In respect to mechanical tests, the steeper decrease in the tangent modulus at 37 °C was related not only to thermal energy, but also to ion exchange, since the PBS bath contained different inorganic salts, where Na^+^ ions participate in the ion exchange with the Ca^2+^ of the hydrogel (Ca^2+^/Na^+^ exchange) [47,48,49,50], which caused its partial decrosslinking. An increase in bath temperature will induce an acceleration in the exchange of these ions and thus, in the degradation of the hydrogel [49]. In addition, there could be a possible release of HA into the medium as it was uncrosslinked. If shear modulus and tangent modulus are compared, the decrease in the modulus was higher for the tangent modulus, especially at higher alginate concentrations where the modulus dropped by ≈40%; however, the shear modulus only reduced its value by ≈10%. The fact that both thermal energy and ion exchange were present in mechanical tests may have contributed to the steeper decrease in the modulus obtained.

For the study of the influence of Alg and HA concentrations in viscoelastic and mechanical properties, the storage modulus *G’* (A) and tangent modulus at 10% of strain (B) of all hydrogels at 37 °C are displayed in Figure 3. Adding HA to the formulation revealed that the storage modulus *G’* and tangent modulus decreased, especially when 2% (*w*/*w*) HA was added (white data, Figure 3). This is the result of the interaction between carboxyl groups of Alg (G-blocks) and amide groups of HA, as demonstrated by FTIR and rheological analysis by several authors [40,41,42,51], which hinders the gel formation by Ca^2+^ chelation since there is a reduction in the available crosslinking points [52,53,54] (see Figure 2B). Consequently, a minor crosslinking density was obtained in the Alg chains and thus, a lower modulus. Regarding the effect of Alg concentration, results show that a higher storage modulus *G’* and tangent modulus were obtained when Alg concentration increased in the formulation due to a higher crosslinking density where more crosslinking points could be found in Alg polymer chains. 

According to the literature, articular cartilage presents a dynamic modulus of 100–3000 kPa [55], matching that of most hydrogels developed in this study, with the group with 2% Alg being the exception since they showed lower values of the modulus (<100 kPa). Further, almost all hydrogels possessed a tangent modulus > 10 kPa, which is in accordance with Naranda et al. [55] who mentioned that natural scaffolds used in cartilage tissue engineering (CTE) exhibit compressive Young’s moduli values in the range of 10–250 kPa, corresponding to very low mechanical properties compared to native cartilage tissue (240–1000 kPa [56,57]). 

### 3.3. Printability Evaluation

The qualitative printability evaluation was performed with all developed hydrogels and was divided into three groups by Alg concentration. So, the influence of HA addition in the printability of hydrogels was studied. All hydrogel solutions showed suitable viscosity (*η* = 1–23 Pas) for 3D bioprinting according to He et al. [58], who remarked that the appropriate viscosity range for a good processability is 0.3–100 Pas. The 4Alg2HA hydrogel was the exception since it presented a very high viscosity (>100 Pas) as mentioned before in Section 3.1, showing a viscosity value out of range, and therefore, making it unsuitable for 3D bioprinting. Consequently, this hybrid hydrogel was rejected for further studies. It should be noted that printability is not only determined by the viscosity of hydrogels because the surface tension of the printing bed plays a crucial role in the quality of printed constructs as well [9,59,60].

Figure 4 shows the different imprint structures fabricated using a needle inner diameter of 27G. A general overview revealed that the addition of HA had a great effect on printability due to the higher ability of the hydrogel precursor to retain the shape of the structure. When HA was added to the formulation, the viscosity of the hydrogel increased, preventing the filament from collapsing. Accordingly, the formulations containing pure Alg in all groups displayed a lower printability than their homologues, causing a slight collapse of the filament and obtaining more irregular and less defined round-like pores, mainly with 2Alg and 3Alg samples. Formulations containing HA, however, presented better printability as a result of their higher viscosity that helped to maintain the shape of the structures. Consequently, good-shaped scaffolds were fabricated with square-like uniform pores. The printability of the hydrogels was enhanced at higher HA concentration as can be observed in the 2Alg2HA and 3Alg2HA samples. 

The suitable viscosity for 3D bioprinting proposed in our previous work by Gorroñogoitia et al. [11] was set to the range of 10–80 Pas, but after the results obtained with the new hybrid formulations developed, the viscosity range could be further adjusted to 1–80 Pas for a good processability or even to 5–80 Pas for a more accurate adjustment, thanks to the interaction between the two biopolymers. 

### 3.4. In Vitro Citotoxicity

The in vitro biocompatibility of hydrogels was evaluated by short–term cytotoxicity assays (Figure 5). As mentioned, the cell viability was measured by incubating L929 murine fibroblast cells for 24, 48, and 72 h in hydrogel extract medium. The results show that none of the developed hydrogels demonstrated signs of cytotoxicity since they all presented cell viability values above the 70% of acceptance limit set down by ISO 10993-5:2009 standard [32] throughout the entire experiment. Furthermore, after 24 h the cell viability increased above 100%, reaching values up to 120% and values of 90% or higher were obtained after 72 h of cell cultivation (Figure 5). 

It is well known that cells are adhered by calcium bonds; therefore, calcium promotes the intercellular interaction. At the same time, calcium is involved in many cellular activities such as cell proliferation and provides a more favourable environment for cell survival [61,62,63,64]. Consequently, the calcium ions that could have been released from the Alg/HA hydrogel to the surrounding medium during the preparation of the liquid extract could be inducing the proliferation of cells and thus, higher cell viability was obtained in the first 24 h. In addition, Herrero-Mendez et al. [19,65] demonstrated that the sulphated glycosaminoglycans (sGAGs), which are known for having specific binding sites for growth factors and cell molecules that are involved in cell adhesion and proliferation [13,17,19], can promote an acceleration in the proliferative rate of cells. Just like calcium, these sGAG molecules could have been liberated to the medium since they remained uncrosslinked within the hydrogels, and consequently, they might be contributing to the higher cell viability obtained at 24 h. 

### 3.5. Cell Morphology and Adhesion

The study of the cells’ morphology and adhesion was performed by the scanning electron microscopy (SEM) analysis that is depicted in Figure 6. As can be observed, in samples containing 2, 3, and 4% alginate (Figure 6A–C), cells were not able to acquire flattened morphology, nor establish cytoplasmic projections adhering to the surface of samples that contained Alg. Accordingly, cells adopted a spherical shape, forming cell aggregates, which was the result of a strong cell–cell interaction rather than cell–matrix interaction, as reported previously by other authors. It is already known that Alg, despite being biocompatible, lacks cell-interactive domains, making it a non-adhesive biomaterial since cells do not have any attachment point with the material itself. To overcome this intrinsic characteristic, Alg can be chemically modified with cell-specific binding motifs such as the RGD peptide that is used as a ligand to promote cell adhesion [24,25,66,67]. Eventually it can also be blended with other materials to induce cell adhesion. For example, Sarker et al. [68] used a hybrid hydrogel composed of alginate and gelatine that was capable of adhering to NHDF cells after 7 days of incubation, unlike a pure alginate hydrogel.

Regarding hydrogels containing HA, cells acquired more flattened morphology than single-alginate hydrogels due to cell–matrix interaction that corroborated the adhesion of cells into the material surface (Figure 6D–H). For example, this was not achieved by several authors [69,70,71] who used a thiol-modified HA or a non-sulphated HA in combination with other biopolymers and the cell adhesion experiments revealed that cells were not able to adhere to the hydrogel surface, acquiring a spherical morphology rather than spindle-like one. The results obtained in this study can be attributed to the ability of sGAGs to bind (non-covalently) through sulphate groups (negatively charged) to growth factors and cell molecules that are involved in many biological activities and functions such as cell proliferation, differentiation, and cell–cell and cell–matrix communication and signalling [13,17,19]. This is reflected in the work of Herrero-Mendez et al. [65] who revealed that adipose-derived stem cells (ASC) and fibroblasts were able to adhere to the hydrogel (HA-enriched + sGAG enriched) and were even viable and capable of repairing chondral and dermal defects by the synthesis of Col-II and Col-I/Fibronectin in vitro, respectively.

### 3.6. Cell Viability of Seeded Cells on Hydrogels and Bioprinted Cells

The viability of murine fibroblasts (L929) after seeding on the surface of hydrogels was examined by live/dead fluorescence staining, which is represented in Figure 7. Most of the cells seeded in all hydrogels were seen to remain viable (stained green) after 7 days of incubation with intact cell membranes and only a few of them died (stained red). The following are the cell viability values for all samples % 91 ± 5 (2Alg), % 91 ± 7 (2Alg1HA), % 94 ± 3 (2Alg2HA), % 86 ± 10 (3Alg), % 90 ± 3 (3Alg1HA), % 88 ± 7 (3Alg2HA), and % 90 ± 10 (4Alg1HA). As also observed by SEM images, cells were found to be agglomerated and forming clusters on Alg single hydrogels (Figure 7A–C) confirming that cell–cell interactions were stronger than cell–material interactions. Nevertheless, this was not the case for Alg/HA hybrid hydrogels, where cells were able to adhere to the material and even proliferate after 7 days of incubation. Furthermore, 7 days after cultivation, cells covered most of the Alg/HA hybrid hydrogel surface (Figure 7D–H); in contrast to Alg single hydrogels where very few cells remain attached, especially in the 3 and 4% alginate hydrogels (Figure 7B,C). Accordingly, it could be concluded that the addition of HA in the formulation of hydrogels, in terms of bioactivity, was favourable for all single Alg hydrogels but mainly for the 3 and 4% alginate ones. Finally, the higher the HA concentration, the greater the cell density on the surface of the hydrogel due to a higher availability of cell-binding sites, thus, increasing the number of adherent cells, as can be observed in Figure 7G,H.

For studying the suitability of the developed hydrogels as precursors for bioprinting, Alg2HA hydrogel was selected because its rheological behaviour and its printability had demonstrated that it could be a good candidate for this purpose. In addition, it showed good cell viability and adhesion. As aforementioned, the bioink was printed using a BIO V1 3D bioprinter with a printing conical nozzle of 22G inner diameter and a flow rate of 2 µL/s, at 37 °C. The cell viability of encapsulated cells in bioprinted constructs was assessed by staining live (green) and dead (red) cells with calcein-AM and propidium iodide. Figure 8 shows the fluorescent images of all samples, both the ones that were not printed (control samples, Figure 8A) and the printed samples (Figure 8B), 1, 3 and 7 days after culture. It can be observed that cells were homogeneously distributed along the constructs resulting in a satisfactory mixing of the cells in the 3Alg2HA hydrogel, respectively. The results obtained show that cells remained viable (stained green) after printing, concluding that the extrusion process did not negatively affect cell viability and thus, the suitability of the 3Alg2HA hydrogel precursor as a good candidate for bioprinting was confirmed due to its appropriate viscosity with shear-thinning properties (Figure 1B, blue data) that avoided cell death. The cell viability remained high during the experiment, showing values for control samples and bioprinted samples of % 77 and % 72 at day 1, % 89 and % 94 at day 3, and % 79 and % 92 at day 7. Apparently, the L929 cells encapsulated in the 3Alg2HA hydrogel did not proliferate over time, showing a constant density of live cells throughout the incubation time.

Although the cell viability was good after printing, it is important to note that during the culturing period of the constructs in MEM, the uncrosslinked HA could have been released to the medium, causing the partial destruction of the scaffold. However, to fully verify that HA is liberated from the hydrogel network, degradation experiments and the quantification of HA should be performed. Moreover, if cells create their own extracellular matrix over time and its synthesis is synchronized to the degradation rate of the hydrogel, there should not be a problem. Anyway, one of the solutions to avoid HA leaching out of the hydrogel network is to crosslink it and obtain IPN hydrogels, instead. However, this was not the aim of this work.

## 4. Discussion and Conclusions

Some research studies have proved the suitability of alginate (Alg) and hyaluronic acid (HA) as mixed biomaterials for cartilage tissue application, as mentioned in the introduction. However, with the aim of adding value and novelty to this investigation, it is important to note that those studies used a different HA from the one used in this work. The HA used in this study differs in composition from the rest of the investigations since it contains a high percentage of sulphated glycosaminoglycans (sGAGs), enhancing the bioactivity properties of the final hydrogels, and at the same time, it allows us to obtain a higher resemblance of the extracellular matrix (ECM) of the native tissue because they are components of the cartilage itself. In addition, the final properties of alginate hydrogels are determined by its microstructure as reported by several authors [11,24,26,27,28]; at the same time, the mechanical properties affect the biological behaviour of cells, so it is very important to have the control of the microstructure of alginate. The alginate used in this work was selected based on the research of our previous work [11], enabling us to develop robust hydrogels.

The semi-interpenetrating (semi-IPN) hydrogels composed of Alg and HA that were developed in this study demonstrated that the final properties of hydrogels could be tailored by varying the concentration of both biopolymers, always seeking a trade-off between mechanical and biological properties. HA was responsible for promoting key biological properties in hydrogels while the presence of Alg was favourable for mechanical properties and hydrogel stability. Except for the 4Alg2HA hybrid hydrogel, which showed too high of a viscosity, the rest of the hydrogels exhibited a suitable viscosity for 3D bioprinting, making them good candidates for bioink applications. In addition, the development of hybrid hydrogels by the combination of Alg and HA, demonstrated a synergetic effect in viscosity but not in mechanical and viscoelastic properties. According to printability results, the stability and quality of printed scaffolds was enhanced by the interaction of the two biopolymers compared to single-alginate scaffolds. The in vitro cytotoxicity assay showed that hydrogels were biocompatible and cell viability experiments by live/dead staining revealed that seeded cells on the surface of hydrogels and encapsulated cells in the printed constructs after bioprinting were viable after 7 days of incubation. Finally, cell adhesion was improved by the incorporation of HA in the formulation. Despite the great potential of all hydrogels in tissue engineering (TE) applications and regenerative medicine, values of the compressive modulus of the hydrogels were far from those of native articular cartilage; the hybrid hydrogels showed a tangent modulus between 3 and 16 kPa, which is still considerably lower than that of native articular cartilage (240–1000 kPa). However, this problem could be addressed either by developing interpenetrating (IPN) hydrogels [72,73,74,75] via the crosslinking of both biomaterials or by adding nanoparticles such as cellulose nanocrystals (CNC) to the formulation [76,77,78,79,80], in order to reinforce mechanically the aforementioned hydrogels. Eventually, some other studies suggest using a high *M_w_* alginate along with a low *M_w_* alginate for the improvement of the mechanical properties [27,81]. Finally, further degradation experiments to analyse the degradation rate of the hydrogels and experiments using chondrocytes or mesenchymal stem cells (MSC) should be carried out so that a 3D cartilage model could be developed by 3D bioprinting. Once this is accomplished, the suitability of these hydrogels in the field of cartilage tissue engineering (CTE) could be confirmed.

## Figures and Tables

**Figure 1 polymers-17-00528-f001:**
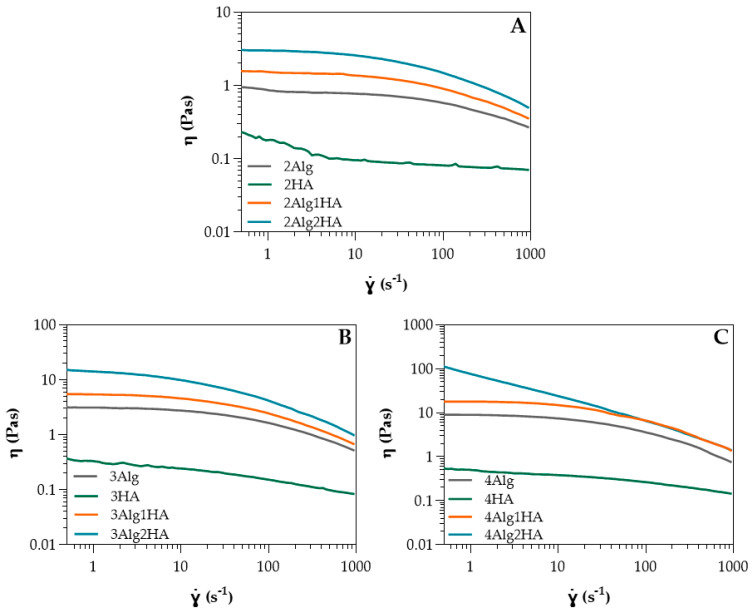
Shear viscosity of HA and Alg solutions at (**A**) 2% (*w*/*w*) and its homologues, (**B**) 3% (*w*/*w*) and its homologues, and (**C**) 4% (*w*/*w*) and its homologues at 37 °C (*n* = 3 per group).

**Figure 2 polymers-17-00528-f002:**
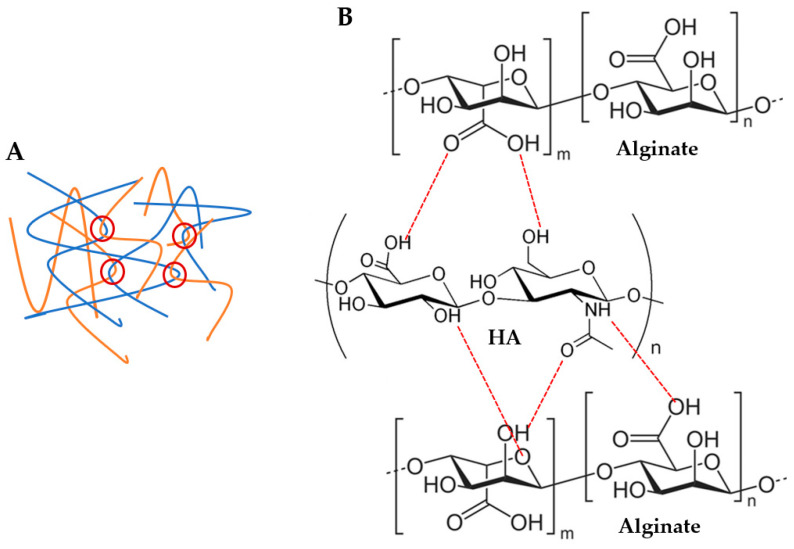
Possible physical interactions between alginate and hyaluronic acid (HA) polymer chains: (**A**) polymer chain entanglements (red circles) and (**B**) H bonds (red dashed lines).

**Figure 3 polymers-17-00528-f003:**
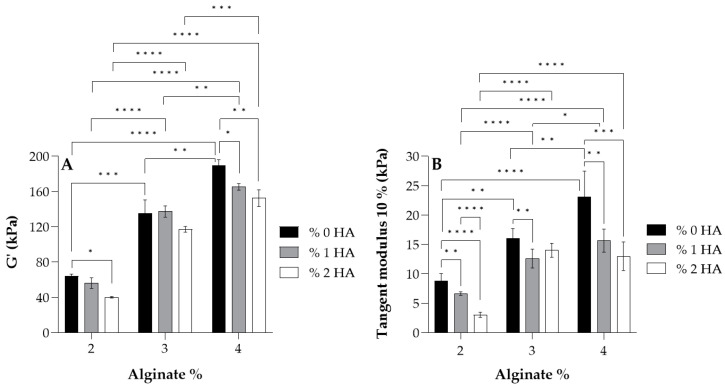
Values of the storage modulus *G’* (**A**) and tangent modulus at 10% of strain (**B**) for each alginate and its homologues crosslinked with 100 mM of CaCl_2_ at 37 °C. Symbols denote statistically significant differences (* *p* < 0.05; ** *p* < 0.01; *** *p* < 0.001; and **** *p* < 0.0001).

**Figure 4 polymers-17-00528-f004:**
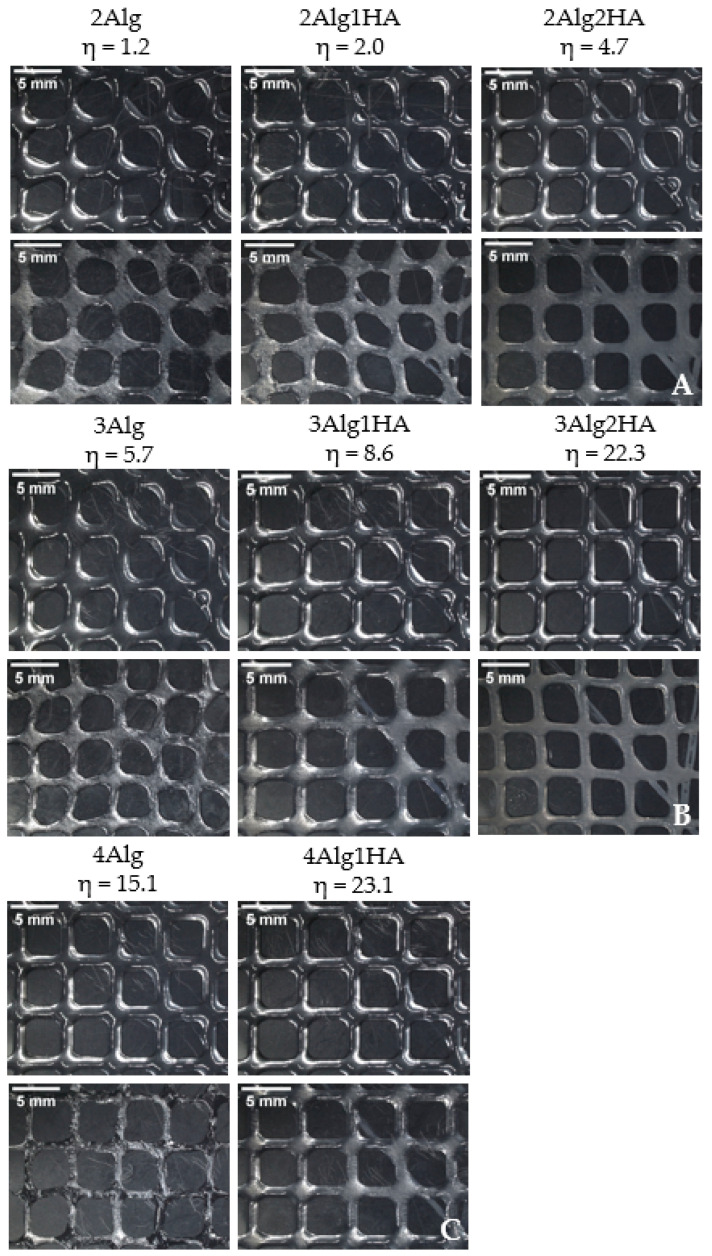
Printed scaffolds of alginate samples and their homologues. Upline: uncrosslinked samples and downline: crosslinked samples. (**A**) Group of 2% (*w*/*w*) alginate (**B**) 3% (*w*/*w*) alginate (**C**) 4% (*w*/*w*) alginate. (Magnification: 0.61×. Needle inner diameter: 27G).

**Figure 5 polymers-17-00528-f005:**
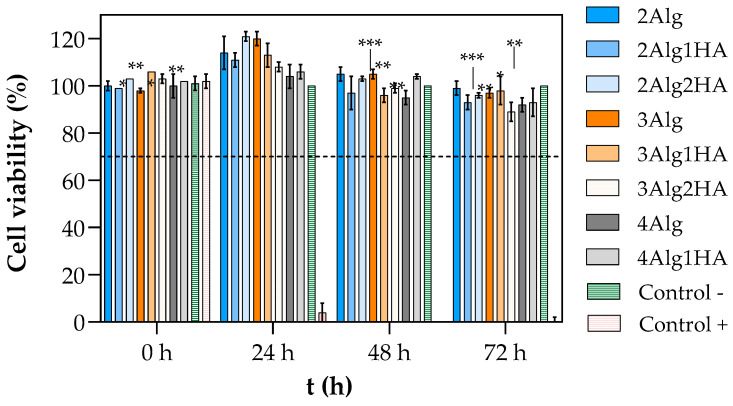
Viability of L929 cells treated with extract liquid of hydrogels crosslinked with 100 mM of CaCl_2_: group of 2% (*w*/*w*) alginate (blue), 3% (*w*/*w*) alginate (orange), 4% (*w*/*w*) alginate (grey), negative control (complete medium with cells, green), and positive control (10% *v*/*v* of DMSO in complete medium, red). The dashed line represents the acceptance limit established by ISO 10993-5:2009 [32] (70% of the negative control value). Symbols denote statistically significant differences in comparison to 24 h (* *p* < 0.05; ** *p* < 0.01; and *** *p* < 0.001).

**Figure 6 polymers-17-00528-f006:**
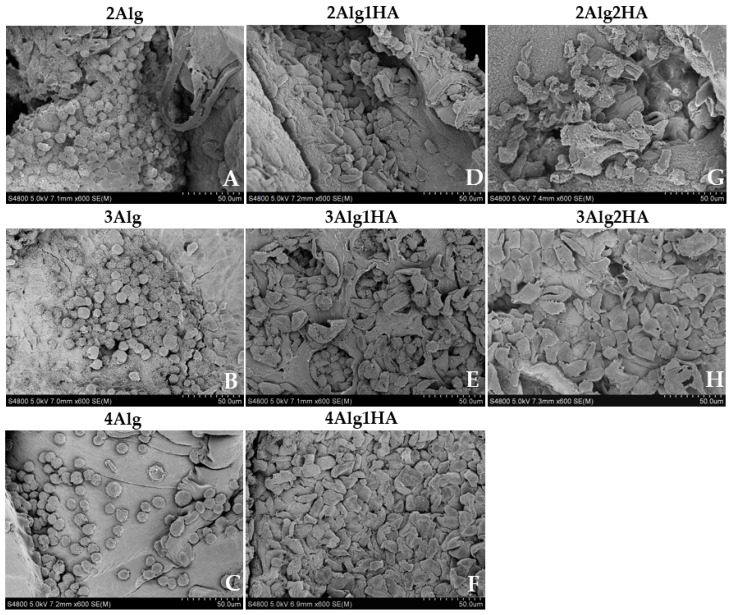
Morphological assessment of L929 cells on alginate (Alg) hydrogels and their homologues crosslinked with 100 mM of CaCl_2_ after 3 days of incubation. Top line: 2% (*w*/*w*) alginate and homologues (**A**,**D**,**G**), middle line: 3% (*w*/*w*) alginate and homologues (**B**,**E**,**H**), and bottom line: 4% (*w*/*w*) alginate and homologues (**C**,**F**) (magnification: 600×).

**Figure 7 polymers-17-00528-f007:**
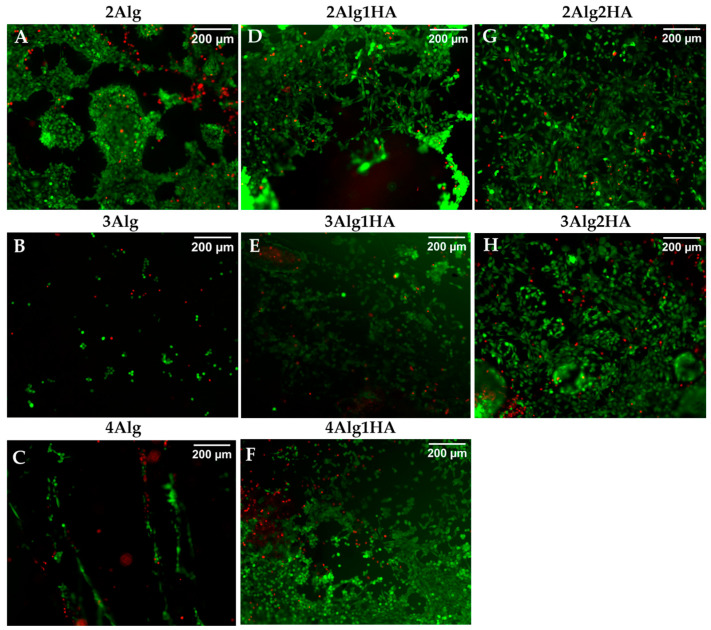
Live/dead cell viability staining of L929 cells on the surface of alginate (Alg) hydrogels and their homologues crosslinked with 100 mM of CaCl_2_ after 7 days of incubation. Top line: 2% (*w*/*w*) alginate and homologues (**A**,**D**,**G**), middle line: 3% (*w*/*w*) alginate and homologues (**B**,**E**,**H**), and bottom line: 4% (*w*/*w*) alginate and homologues (**C**,**F**). Dead cells are stained red and live cells are stained green (magnification: 10×).

**Figure 8 polymers-17-00528-f008:**
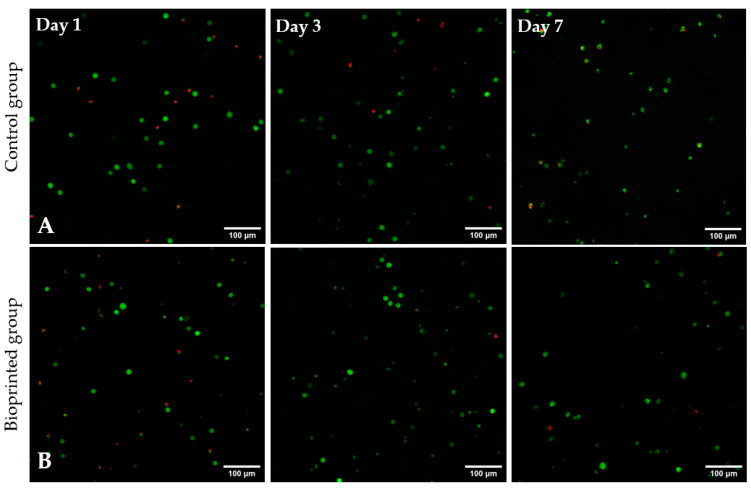
Live/dead images of L929 cells encapsulated in the 3Alg2HA hydrogel after 1, 3, and 7 days of incubation: (**A**) control samples and (**B**) bioprinted constructs. Dead cells are stained red and live cells are stained green (magnification: 10×).

**Table 1 polymers-17-00528-t001:** Native whartonide hyaluronic acid properties.

Designation	Source	Batch	*M_w_* (kDa)	^a^ sGAGs
HA	Wharton’s jelly	210,041	<2000	Yes
		230,002 + 230,005		
		230,029 + 230,056		

^a^ Sulphated glycosaminoglycans.

**Table 2 polymers-17-00528-t002:** Sodium alginate properties. *M_w_* obtained from gel permeation chromatography (GPC) (polyethylene glycol (PEO) standards).

Designation	Source	Batch	*M_w_* (kDa)
Alg	Brown algae	BCCD8789	392

**Table 3 polymers-17-00528-t003:** Alg/HA acid formulations obtained from the mixtures at different proportions in PBS.

Formulation	Alginate % (*w*/*w*)	HA % (*w*/*w*)	Molar Ratio Alg/HA
2Alg1HA	2	1	10:1
2Alg2HA	2	2	5:1
3Alg1HA	3	1	15:1
3Alg2HA	3	2	7.6:1
4Alg1HA	4	1	20:1
4Alg2HA	4	2	10:1

## Data Availability

The original contributions presented in this study are included in the article/Appendix A. Further inquiries can be directed to the corresponding author.

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
