# Peer review of "The Effect of Alginate/Hyaluronic Acid Proportion on Semi-Interpenetrating Hydrogel Properties for Articular Cartilage Tissue Engineering†"

_polymers, 2025, doi:10.3390/polym17040528_

Round 1

Reviewer 1 Report

Comments and Suggestions for Authors

While this study is nicely presented, its novelty could be questioned. I understand that the novelty of this paper amongst the myriad of other papers on alginate/HA  focuses on the specificity of the HA, but I am unsure if this point alone certifies its novelty.

As such, additional analysis could have been performed to improve this paper such as:

1) Including recovery rheological tests that mimic the 3D printing process, especially considering that 3D printing was a key factor in this study. The authors could refer to:  Synthesis and evaluation of alginate, gelatin, and hyaluronic acid hybrid hydrogels for tissue engineering applications, A Serafin, M Culebras, MN Collins, International journal of biological macromolecules 233, 123438

2) Swelling and degradation tests could also have been included, in particular knowing that the physical crosslinking of Alg by means of CaCl2 is easily reversible. The authors' intended target tissue of cartilage, a tissue that is very hard and slow to regenerate, would prompt the question of whether these developed materials would survive long enough in-situ to prompt any regenerative benefit to the patient.

3) Quantification of the live/dead image data would more strongly support the statements of viability made by the authors. The same could be said of the SEM/cell images.

4) The intended choice of tissue (cartilage) feels like an afterthought in this paper. I would suggest either the authors remove this and focus this paper on more broad material characterisation or provide more details about why these particular materials are suitable for cartilage tissue engineering repair. 

Author Response

Comments 1: Including recovery rheological tests that mimic the 3D printing process, especially considering that 3D printing was a key factor in this study. The authors could refer to: Synthesis and evaluation of alginate, gelatin, and hyaluronic acid hybrid hydrogels for tissue engineering applications, A Serafin, M Culebras, MN Collins, International journal of biological macromolecules 233, 123438

Response 1: Thank you for pointing this out. We have added the reference to the manuscript. You may find it in line 270 of Section 3.1.

Comments 2: Swelling and degradation tests could also have been included, in particular knowing that the physical crosslinking of Alg by means of CaCl2 is easily reversible. The authors' intended target tissue of cartilage, a tissue that is very hard and slow to regenerate, would prompt the question of whether these developed materials would survive long enough in-situ to prompt any regenerative benefit to the patient.

Response 2: Thank you very much for the comment. We recognize the importance of including swelling and degradation tests, especially considering the reversible nature of the physical crosslinking of Alg through CaCl2 and the aim of regenerating cartilaginous tissue. Unfortunately, due to time limitations, we cannot conduct these additional experiments before the submission deadline for this manuscript. However, we plan to include these tests in future studies to fully approach this aspect and provide more complete data about the long-term degradation of the developed hydrogels.

Comments 3: Quantification of the live/dead image data would more strongly support the statements of viability made by the authors. The same could be said of the SEM/cell images.

Response 3: Thank you very much for the comment. Regarding the LIVE/DEAD samples, we have quantified them using the color threshold and particle count techniques with ImageJ, performing different sections in the images. The images were divided into four sections, the quantification was carried out, and the statistical calculation was conducted. You may find detailed information in lines 246-249 of the manuscript, in Section 2.9. The results have been included in Section 3.6 of the manuscript in lines 475-478 and 512-514.

In regard with the SEM samples, the quantification could not be performed due to the magnification employed. The magnification offered by this technique is so high that the number of cells that can be observed is too small to obtain an accurate quantification. These measures were applied to explore the cellular morphology, in order to determine whether the cells adhered to the material, as well as to analyze the attachment points between cells.

Comments 4: The intended choice of tissue (cartilage) feels like an afterthought in this paper. I would suggest either the authors remove this and focus this paper on more broad material characterisation or provide more details about why these particular materials are suitable for cartilage tissue engineering repair. 

Response 4: Thank you very much for pointing this out. I would to like to clarify that this work is the continuation of a previous study about alginate hydrogels (https://doi.org/10.3390/polym14020354) and is part of a doctoral thesis in which the final application is joint cartilage. The reason for selecting these specific materials is detailed in the Introduction of the manuscript, in lines 53-81.

Reviewer 2 Report

Comments and Suggestions for Authors

The authors have put a strong emphasis on experimental selections for their project but there are some major issues which need to be addressed properly.

1.      In section 2.9, please explain the experimental section with clarity. Is the control set being referred to cells cultured on the surface of the hydrogel? Is this compared to the cells seeded in the hydrogel constructs?

2.      The results from Live/dead assay must be in a quantified format to have a better understanding of the represented data. The figure showing the live/dead count in control and 3Ag2HA does not show many differences which is a positive fact, but the data needs to be represented in a quantified format.

3.      What is the rational behind showing cell morphology on the surface rather than the morphology with the bio-printed hydrogels here in section 2.8?

4.      The authors must put more stress on defining the need for “semi-interpenetrating (Semi-IPN) hydrogels” used in their application from various relevant resources and talk about the selection criteria for the particular kind more elaborately in the introduction to justify the title of the research article. Authors could have also selected self-degradable/sacrificial hydrogels which promote chondrocyte binding and penetration within having similar mechanical properties. This could have naturally mimicked collagen formation due to attachment of chondrocytes as the hydrogel degrades over time. Justification statement on the initial hydrogel base material type must be more elaborate.

5.      Please provide higher resolution images with proper background correction in Figure 7. Some of the panels show very high fluorescence from the background, which is not preferable here.

Comments on the Quality of English Language

Some of the sentences are too long which can be broken down to keep the readers' interest flowing.

Example: For this, we used a HA that came from natural origin (Wharton’s jelly extracted from the umbilical cord) and contained high amount of sGAGs, which allowed us to develop bioactive hydrogels since these sGAGs acted as bioactive molecules, providing cell adhesion and proliferation.

Rectified format: We used a HA that came from natural origin (Wharton’s jelly extracted from the umbilical cord) and contained a high amount of sGAGs. This helped us to develop bioactive hydrogels as sGAGs acted as bioactive molecules for providing cell adhesion and proliferation.

Author Response

Comments 1: In section 2.9, please explain the experimental section with clarity. Is the control set being referred to cells cultured on the surface of the hydrogel? Is this compared to the cells seeded in the hydrogel constructs?

Response 1: Thank you for the comment. In this section, on the one hand, we analyze the cells seeded on the hydrogel, which correspond to the first paragraph of section 2.9 and Figure 7 of the manuscript. On the other hand, in the second paragraph of Section 2.9, we analyze the cells embedded in the hydrogel, which correspond to the 3D-bioprinted cells (Figure 8). In this second paragraph, the control samples refer to the samples that were not 3D-bioprinted in order to examine the influence of bioprinting on cell viability. You may also find this information in line 197 of the manuscript, in Section 2.6. I hope I managed to clarify your doubts.

Comments 2: The results from Live/dead assay must be in a quantified format to have a better understanding of the represented data. The figure showing the live/dead count in control and 3Ag2HA does not show many differences which is a positive fact, but the data needs to be represented in a quantified format.

Response 2: Thank you very much for pointing this out. We have added the quantification of the LIVE/DEAD results of Figure 8 in Section 3.6 of the manuscript in lines 512-514. The quantification was performed by means of the color threshold and particle count techniques with ImageJ, which is detailed in Section 2.9 of the manuscript.

Comments 3: What is the rational behind showing cell morphology on the surface rather than the morphology with the bio-printed hydrogels here in section 2.8?

Response 3: Thank you very much for the comment. The SEM was conducted to analyze the adhesion of the cells to the material, in order to verify whether the cells employ some interaction with the material (attachment points), which would determine their morphology. When the cells are encapsulated within the material (corresponding to the bioprinted hydrogels), the technique we should use to analyze the morphology is confocal microscopy, although it is not as precise as SEM in this regard. SEM is basically employed to analyze surfaces, since the penetration capacity of the electron beam is very limited.

Comments 4: The authors must put more stress on defining the need for “semi-interpenetrating (Semi-IPN) hydrogels” used in their application from various relevant resources and talk about the selection criteria for the particular kind more elaborately in the introduction to justify the title of the research article. Authors could have also selected self-degradable/sacrificial hydrogels which promote chondrocyte binding and penetration within having similar mechanical properties. This could have naturally mimicked collagen formation due to attachment of chondrocytes as the hydrogel degrades over time. Justification statement on the initial hydrogel base material type must be more elaborate.

Response 4: Thank you very much for pointing this out. I would like to clarify that this work is the continuation of a previous study about alginate hydrogels (https://doi.org/10.3390/polym14020354) and is part of a doctoral thesis in which the final application is joint cartilage. The reason for selecting these specific materials is detailed in the Introduction of the manuscript, in lines 53-81.

Moreover, the goal we set from the beginning in this work was to develop semi-IPN hydrogels, where only one of the materials is cross-linked. The rheological, viscoelastic and biological study of these hydrogels was too extensive, and thus we could not include interpenetrating (IPN) hydrogels. In the future, our objective is to develop IPN hydrogels in which both materials are cross-linked. We will also carry out the bioprinting with chondrocytes to confirm that these materials are appropriate for the regeneration of joint cartilage.

Comments 5: Please provide higher resolution images with proper background correction in Figure 7. Some of the panels show very high fluorescence from the background, which is not preferable here.

Response 5: Thank you very much for the comment. Accordingly, Figure 7 has been modified.

Reviewer 3 Report

Comments and Suggestions for Authors

1.     The manuscriptThe effect of Alginate/Hyaluronic acid proportion on semi-interpenetrating hydrogel properties for Articular Cartilage Tissue Engineering”, covers an interesting topic of composite scaffolding in soft tissue engineering.

The manuscript requires a significant major revisions.

Introduction should focus on the novelty of the scaffold fabricated. How it is novel in respect to other articles on Alginate/HA scaffolds fabricated by bioprinting.

2.      In the methodology, the authors described that only Alginate crosslinked with the calcium chloride and not HA. Could any other crosslinker  be used so that Alg/HA crosslinked network was possible. This might have improved the cell viability, adhesion and proliferation too. Kindly justify the free presence of HA in the scaffold.

3.      Line: 402, 468: (Figure 5Error! Reference source not found.). Correct it.

4.      Figure 6. Cell morphology over the scaffolds do not reveal clear spindle-shaped cell growth and ability to proliferation (which must be visible in SEM image, if the scaffold has supported cellular growth). Mostly rounded cells were seen that indicate no growth and proliferation. The authors are required to show Oval or spindle shaped L929 cells (signify healthy and proliferating L929 cells ) marked with arrow in the FESEM images. Otherwise, the authors should not mention that the cells are proliferating well on scaffold (Line 413).

5.      Control is not mentioned in the MTT experiment Figure 5. It should be included in figure itself.

6.      Line 413: Alg/HA hydrogel to the surrounding medium during the preparation of the liquid extract,  could be inducing the proliferation of cells and thus, higher cell viability was obtained in the first 24 h. MMT results are showing some viable cells. However, the MTT results are not correlating with FESEM images. It could not be claimed that cells are proliferating over the scaffold.

Author Response

Comments 1: Introduction should focus on the novelty of the scaffold fabricated. How it is novel in respect to other articles on Alginate/HA scaffolds fabricated by bioprinting.

Response 1: Thank you very much for the comment. The novelty of the developed hydrogels is described in the Discussion and Conclusions sections, in lines 531-544 of the manuscript. In this work, we stress on the intrinsic properties of the starting materials, which differ from those of the rest of the studies conducted to date and determine the final properties of the hydrogels.

Comments 2: In the methodology, the authors described that only Alginate crosslinked with the calcium chloride and not HA. Could any other crosslinker  be used so that Alg/HA crosslinked network was possible. This might have improved the cell viability, adhesion and proliferation too. Kindly justify the free presence of HA in the scaffold.

Response 2: Thank you very much for pointing this out. The goal we set from the beginning in this work was to develop semi-IPN hydrogels based on alginate and hyaluronic acid. The rheological, viscoelastic and biological study of these hydrogels was too extensive, and thus we could not include interpenetrating (IPN) hydrogels. In the future, our objective is to develop IPN hydrogels in which both materials are cross-linked. We will also carry out the bioprinting with chondrocytes to confirm that these materials are appropriate for the regeneration of joint cartilage.

Comments 3: Line: 402, 468: (Figure 5Error! Reference source not found.). Correct it.

Response 3: Thank you very much for your thorough revision. The errors have been corrected.

Comments 4: Figure 6. Cell morphology over the scaffolds do not reveal clear spindle-shaped cell growth and ability to proliferation (which must be visible in SEM image, if the scaffold has supported cellular growth). Mostly rounded cells were seen that indicate no growth and proliferation. The authors are required to show Oval or spindle shaped L929 cells (signify healthy and proliferating L929 cells ) marked with arrow in the FESEM images. Otherwise, the authors should not mention that the cells are proliferating well on scaffold (Line 413).

Response 4: Thank you very much for the comment. We totally agree that the SEM images do not show a clear shape of the proliferated cells. However, the comparison between the samples that contained hyaluronic acid (HA) and those that did not contain HA showed a difference in cell morphology, even though they had not fully proliferated. With this result, we concluded that the addition of HA improves cell adhesion, although it is not optimal. Furthermore, in Figure 7, which corresponds to the LIVE/DEAD assays, the cell density of the samples that contained HA was much greater than that of the samples that only contained alginate, which indicates that the cells adhered to the material and proliferated in the presence of HA. This is the result we aimed to achieve with the addition of HA to alginate.

Comments 5: Control is not mentioned in the MTT experiment Figure 5. It should be included in figure itself.

Response 5: Thank you very much for the comment. We have, accordingly, added the controls to Figure 5.

Comments 6: Line 413: Alg/HA hydrogel to the surrounding medium during the preparation of the liquid extract,  could be inducing the proliferation of cells and thus, higher cell viability was obtained in the first 24 h. MMT results are showing some viable cells. However, the MTT results are not correlating with FESEM images. It could not be claimed that cells are proliferating over the scaffold.

Response 6: Thank you very much for the comment. We have modified the text in an attempt to respond to the reviewer’s point, hoping that we understood the meaning of the comment. In this case, these are two different procedures with two different objectives: 1) the in vitro cytotoxicity assays were conducted using the indirect method of hydrogel extractive medium, thus the cells were seeded in a multi-well plate (not on the hydrogel), and we added the extractive medium that was in contact with the hydrogel, in order to analyze the biocompatibility of the material. 2) The SEM assays were conducted with cells seeded on the surface of the hydrogel. The aim of this assay was to analyze the morphology that the cells adopted, as well as the attachment points of the cells to the material or among them. The comparison of both experiments is incompatible.